# Portrait of Cancer Stem Cells on Colorectal Cancer: Molecular Biomarkers, Signaling Pathways and miRNAome

**DOI:** 10.3390/ijms22041603

**Published:** 2021-02-05

**Authors:** Andrea Angius, Antonio Mario Scanu, Caterina Arru, Maria Rosaria Muroni, Vincenzo Rallo, Giulia Deiana, Maria Chiara Ninniri, Ciriaco Carru, Alberto Porcu, Giovanna Pira, Paolo Uva, Paolo Cossu-Rocca, Maria Rosaria De Miglio

**Affiliations:** 1Institute of Genetic and Biomedical Research (IRGB), CNR, Cittadella Universitaria di Cagliari, 09042 Monserrato, Italy; vincenzo.rallo@irgb.cnr.it; 2Department of Medical, Surgical and Experimental Sciences, University of Sassari, Via P. Manzella, 4, 07100 Sassari, Italy; scanu@uniss.it (A.M.S.); mrmuroni@uniss.it (M.R.M.); g.deiana43@studenti.uniss.it (G.D.); m.ninniri@studenti.uniss.it (M.C.N.); alberto@uniss.it (A.P.); rocco@uniss.it (P.C.-R.); 3Department of Biomedical Sciences, University of Sassari, 07100 Sassari, Italy; 30039590@studenti.uniss.it (C.A.); carru@uniss.it (C.C.); gpira@uniss.it (G.P.); 4IRCCS G. Gaslini, 16147 Genoa, Italy; paolouva@gaslini.org; 5Department of Diagnostic Services, “Giovanni Paolo II” Hospital, ASSL Olbia-ATS Sardegna, 07026 Olbia, Italy

**Keywords:** colorectal carcinoma, cancer stem cells, miRNAs, regulatory network, signaling pathways, feedback loop

## Abstract

Colorectal cancer (CRC) is a leading cause of cancer death worldwide, and about 20% is metastatic at diagnosis and untreatable. Increasing evidence suggests that the heterogeneous nature of CRC is related to colorectal cancer stem cells (CCSCs), a small cells population with stemness behaviors and responsible for tumor progression, recurrence, and therapy resistance. Growing knowledge of stem cells (SCs) biology has rapidly improved uncovering the molecular mechanisms and possible crosstalk/feedback loops between signaling pathways that directly influence intestinal homeostasis and tumorigenesis. The generation of CCSCs is probably connected to genetic changes in members of signaling pathways, which control self-renewal and pluripotency in SCs and then establish function and phenotype of CCSCs. Particularly, various deregulated CCSC-related miRNAs have been reported to modulate stemness features, controlling CCSCs functions such as regulation of cell cycle genes expression, epithelial-mesenchymal transition, metastasization, and drug-resistance mechanisms. Primarily, CCSC-related miRNAs work by regulating mainly signal pathways known to be involved in CCSCs biology. This review intends to summarize the epigenetic findings linked to miRNAome in the maintenance and regulation of CCSCs, including their relationships with different signaling pathways, which should help to identify specific diagnostic, prognostic, and predictive biomarkers for CRC, but also develop innovative CCSCs-targeted therapies.

## 1. Introduction

Colorectal cancer (CRC) is the third most common cancer and the second most frequent source of cancer-related mortality worldwide [1]. Arnold et al. recognized three distinct global temporal patterns to CRC development trends: A rise in both incidence and mortality (Baltic countries, Russia, China, and Brazil); an increasing incidence, but decreasing in mortality (Canada, the United Kingdom, Denmark, and Singapore); and reduction in both incidence and mortality (the United States, Japan, and France) [2]. In highly advanced nations, the increased cancer-incidence highlights the influence of dietary habits, obesity, and lifestyle. The reduced cancer-mortality is attributable to population-based screenings and significant progresses in therapeutic options improving CRC patient’s management [2]. Approximately 10% of CRC patients under 55 years showed more severe and unfavorable pathological features than older cohorts, resulting in a negative impact on their survival outcome [3].

CRC is a well-studied malignancy for which extensive and heterogeneous genomic aberrations, well-defined risk factors, slow progression, and identifiable and treatable preneoplastic lesions have been described [4,5]. In patients at stage I of the disease, the five-year survival rate is 90%, but we observed that a drastic reduction of slightly more than 10% is observed when cancer patients reach stage IV [6]. Approximately 20% of CRC patients already have metastases at diagnosis, and metastatic CRC (mCRC) is generally an incurable disease [7]. CRC is a heterogeneous multifactorial disease presenting significant differences in prognoses and responses to treatment. The importance of detecting specific pathway abnormalities is crucial to improve diagnosis, prognosis, and therapeutic strategies. CRC molecular alterations permit us to identify two distinct genetic pathways. The adenoma-carcinoma pathway, defined as chromosomal instability (CIN) responsible for up to 85% of CRC, and the serrated neoplasia pathway, accounting for the remaining 15%. CIN mechanisms involved chromosome alterations and DNA damage response network, affecting critical genes involved in cell function (APC, KRAS, PI3K, TP53) and pathways (WNT, MAPK, PI3K, TGF-β) [8]. The serrated neoplasia pathway is associated with RAS and BRAF gene mutations and epigenetic instability, characterized by the CpG island methylator phenotype (CIMP). Genome-wide studies identified new markers and phenotypic subtypes based on polymerase-mutations or mismatch repair deficiency leading to a hypermutated phenotype [8]. These two latter molecular events explain the microsatellite instability (MSI) identified in the 15–20% of CRCs [9].

Current biomarkers in mCRC treatment decision involve evidence of KRAS and NRAS mutations. A clear clinical meaning has only been achieved by KRAS oncogene in CRC patient management. KRAS oncogene regulates the activation of downstream effectors of several pathways, such as BRAF/RAS/MAPK, PI3K/AKT, RalGDS/p38-MAPK, etc., thus influencing normal cell physiology, neoplastic cell biology, and therapeutic responses. At least 40% of CRCs reported KRAS mutations that are biomarkers predictive of treatment efficacy and patient outcome [10]. Particularly, exon 2 KRAS mutations are correlated to advanced stage tumors and adverse prognosis [11]. Identification of KRAS mutations is a key molecular test for evaluating targeted therapies in mCRC. The presence of wild-type KRAS sequences guarantees the success of targeting by monoclonal antibodies (Cetuximab or Panitumumab) of the EGFR axis [12]. BRAF gene mutation (V600E) has been associated with aggressive clinical outcome in CRC patients [13]. Five percent of mCRC patients show a link between MSI-high (MSI-H) and a striking response to immune checkpoint blockade with anti-PD1 therapy [14]. Moreover, 3–8% of KRAS-wildtype CRC patients exhibit anti-EGFR therapy resistance explained by the presence of HER2 amplification.

Despite extensive knowledge of CRC biology and improvements in therapy, such as surgery, chemotherapy, and targeted therapies, this tumor remains one of the hard-to-treat cancers considering the high frequency of metastases and recurrences after surgery, and frequent resistance to first or second-line of treatment.

Cancer stem cells (CSCs) are multipotent and self-renewing cells whose role was first identified in hematologic malignancies and recently in solid tumors, including CRC [15]. Some evidence supports the hypothesis that onset, progression, and development of drug resistance in CRC might be related to the maintenance of a CSCs phenotype by deregulation of pathways involved in transformation, differentiation, growth, and epithelial to mesenchymal transitions (EMT) [16].

Recent data have validated the significant role of epigenetics in regulating the function of CRC cells [17] and colorectal cancer stem cells (CCSCs) [18]. Non-coding RNAs (ncRNAs) such as microRNAs (miRNAs) and long-non-coding RNAs (lncRNAs) control several cell functions and regulate gene expression by interacting with target mRNAs, resulting in either mRNA degradation or translational repression [19].

The relationship between miRNAs gene expression profile and CRC clinical outcome has been extensively analyzed [20]. MiRNAs control several cellular functions, including self-renewal and cellular differentiation, that make SCs insensitive to environmental stimuli that would normally stop most cells at the G1/S checkpoint. It could be proposed that the mechanisms used by SCs to overcome this checkpoint should be used by tumor cells to progress [21].

This review summarizes the role of CSCs in the CRC pathogenesis, describes the emerging roles of miRNAs in the epigenetic regulation of CCSCs properties, and discusses their potential clinical utility.

## 2. Intestinal Stem Cells Are Key Drivers of Epithelial Homeostasis and Regeneration

The intestinal stem cells (ISCs) are characterized by unlimited self-renewal ability, long cell cycling, and an initial undifferentiated status that has the ability to differentiate into specific cell lineages. They are also capable of genome repair and protection from the microenvironment under attack [22,23]. The intestinal epithelium is the most rapidly renewing tissue in our body. It undergoes a continuous and complete cell turnover (every 4 to 5 days) during entire life [24]. The cells at the villus tip are dispersed in intestinal lumen and constantly replaced by new specialized epithelial cells, a process termed homeostasis. The intestinal epithelium undergoes continuous aggression from the luminal contents: More than 10^11^ epithelial cells are being lost every day in humans [25]. The intestinal SCs, inserted at the base of the crypt of Lieberkühn, are responsible for homeostasis and injury repair processes. We distinguish them in the rapidly cycling intestinal stem cells (ISCs) positive for LGR5 marker, also called crypt base columnar (CBC), located at the crypt base and interspaced between Paneth cells, and the slow-cycling label-retaining HOPX+, LRIG1+, BMI1+, TERT+, DLL1+ ISCs, also called label-retaining cells (LRC), located at the +4-position relative to the crypt base. All of these cells are surrounded by the niche [26,27,28,29,30].

During intestinal homeostasis, the actively LGR5+ ISCs split up symmetrically and stochastically and generate transit-amplifying (TA) cells, which cover the rest of the crypt. TA cells possess a high proliferative and migratory ability and undergo differentiation in short-lived nutrient-absorbing enterocytes [26,31,32]. LGR5+ ISCs can also activate DLL1 expression and commit to the secretory progenitor cells, which supply mucin-secreting goblet cells, peptide hormone-secreting neuroendocrine cells, and tuff cells. This process replaces cells that are lost via anoikis at the villus tip. Paneth cell turnover is the only exception to rapid self-renewal (every 3–6 weeks), generated from dedicated secretory cell progenitors located in the TA compartment, which mature into differentiated defensing-Paneth cells and follow a migratory path to the crypt bottom [31,33,34].

Accumulating evidence has suggested the existence of quiescent SCs in the intestinal epithelium, called reserve SCs [27,28,35,36,37,38,39]. The tissue injury causes a loss of actively proliferating LGR5+ ISCs: A regenerative response needs to recover their pool. Two models have been postulated: The reserve SC model and the dedifferentiation model. The first one considered the reserve ISCs playing their main function in the injured intestinal epithelium regeneration, being resistant to DNA-damaging and able to active a proliferative response increasing CBC lineage in response to such injury [40]. The reserve ISC are considered indispensable during intestinal homeostasis [41] undergoing periodically division to generate CBCs driven by high WNT signaling and marked by LGR5 expression. The second model suggests the capacity of progenitor cells or terminally differentiated cells to enter again in the cell cycle and recover the SC activity, regenerating the damaged epithelium [42,43,44,45,46].

The niche is a term conceived by Schofield in 1978 [47]. It has a complex biochemical composition and biophysical architecture. It provides and integrates a plentiful of local and systemic signals decisive for maintaining SCs functions, especially to ensure appropriate and coordinated SCs responses to support the changing of tissues [48], both in the intestinal epithelium homeostasis and in the regenerative processes of the intestinal epithelium under injury conditions.

The WNT signaling is the main pathway in regulating intestinal epithelial self-renewal and proliferation. The gradient of WNT signaling exists along the crypt-villus axis explaining the progressive loss of proliferative capacity and the acquisition of differentiation of the cells migrated away from the WNT source at the base of the crypt [49]. WNT signaling is responsible for the expression of EphB/EphrinB, which affect the position of the different cells along the crypt-villus axis [50].

An inverse gradient of BMP signaling occurs alongside the crypt-villus axis. High villus activity inhibits cell proliferation and lower activity within the crypt, where a restricted gradient of BMP antagonists originates from stromal cells, supports WNT signaling in ISCs [51]. Notch signaling activity also contributes to the development of an absorptive progenitor and ultimately mature enterocytes [52].

Different feedback signals between SCs and cell progeny have been identified. Paneth cells release important signaling molecules such as WNT3, EGF, and Notch ligand DLL4, which represent an important signal to ISCs [53,54]. Alternative pathways, such as stromal secretion of WNTs, support intestinal homeostasis [55,56,57]. WNT signaling promotes LGR5+ ISC self-renewal by additional signals from R-spondin receptors (LGR4, LGR5, LGR6), which were stabilized and enabled to drive ISC expansion by WNT itself [58]. Furthermore, Paneth cells produce and provide lactate to ISCs, which increase the oxidative phosphorylation with ROS production and differentiation [59]. ISCs send feedback signals to cell progeny, maintaining their terminally differentiation and participating in self-signaling loops. In the intestine, Notch signaling promotes LGR5+ ISCs proliferation and prevents differentiation in secretory cell lineage by tuning local WNT signaling activity. ISCs regulate homeostasis of their daughter cells and control the balance between self-renewal and differentiation [60,61,62]. In the niche, various stromal cells (mesenchymal cells, endothelial cells, immune cells) showed an active role in maintaining ISCs by secreting signaling factors such as WNT, Notch, and BMP [63]. Several studies have proven that intestinal progenitor cells, and even more differentiated cell types, exhibit deep plasticity, giving them the ability to transdifferentiate and dedifferentiate under specific homeostatic and regenerative conditions. These processes have been orchestrated by WNT signaling of the niche [64]. New evidence suggests the role of the architecture of the niche and the mechanical regulation of ISCs positioning, density, contractility, and compression within the niche as strong self-organizing conditions that control ISC functions [64].

Identification of unambiguous intestinal SC-markers and molecular signaling between the cells and stroma, aside from the development “in vitro” and “in vivo” of new technologies, should allow huge progress in knowing how intestinal SCs interact with niche to maintain homeostasis in healthy intestinal epithelia, and how they could contribute to intestinal cancer initiation and progression [65,66,67]. The ISC features make them ideal for regenerative medicine approaches and an excellent candidate for accumulation of mutations that promote cancer growth and therefore attractive as therapeutic targets.

## 3. Cancer Stem Cells in Colorectal Cancer

Recent technological progresses have identified several genetic and epigenetic aberrations explaining the heterogeneous nature of CRC and justifying the requirement of personalized medicine for CRC patients. Additionally, CRC consists of heterogeneous cell populations differing for markers expression, proliferative capacity, morphology, and malignant potential. CRC intra-tumor heterogeneity is the seed of cancer formation due to the presence of CSCs. The context in which the tumor should be studied by applying the principles of stem cell biology is given by the clonal expansion and repopulation characteristics of the various cell lineages of CSCs within the tumor [68].

ISCs and transit-amplifying cells are the best candidates for the origin of CCSCs, given that their high proliferative capacity allows easy acquisition of genetic and epigenetic aberrations. Although, studies on histology of intestine and CRC suggested that CCCSs might arise from terminally differentiated cells [69]. Recent studies on de-differentiation process suggested that intestinal non-stem cell mutated and persevered could lead to adenoma progression in the presence of additional genomic events (e.g., b-catenin mutation combined with increased NFKB signaling or changes in the microenvironment) [70]. ISC features could be sustained by genetic/epigenetic aberrations acquired into different neoplastic subclones [71]. Stemness is responsible for three key processes that trigger tumor progression: Cell growth and proliferation, recurrence and metastasization, and therapy resistance (Figure 1).

A convincing link between EMT and CSCs explained their crucial role in CRC progression and therapeutic resistance [72]. EMT-associated gene expression gives invasive and metastatic characteristics, resistance to therapies, and CSCs phenotypes on cancer cells [73,74,75,76]. The clinical implication is that by removing the CCSC, the tumor will no longer be capable of growing. Based on the CSC hypothesis, our diagnostic and therapeutic goals will be specifically discovered and targeted CCSCs [76].

Colorectal CSCs have been characterized based on specific markers. CD133 is well documented: It is a membrane-bound glycoprotein involved in primordial cell differentiation and EMT [77,78,79,80]. CD133+ cells produce IL-4 and use it to evade apoptosis. The treatment with IL-4-neutralizing antibody or with IL-4Rα antagonist will greatly enhance the sensitivity of CD133+ cells to chemotherapeutic drugs increasing antitumor efficacy, confirming previous observation [81]. CD133+ CRC cells manifested the CSC-like properties, such as higher levels of SC markers OCT4 and SOX2, tumor sphere forming ability, and more tumorigenic in NOD/SCID mice [82], which is consistent with OCT4 and SOX2 overexpression in poorly differentiated human tumors [83]. CSCs display EMT phenotype, possess high levels of transcription factors SNAIL and TWIST, mesenchymal markers vimentin and fibronectin, and low levels of epithelial protein E-cadherin. CD133+ CSCs might escape immune surveillance by expressing inhibitory molecule B7H1. The EMT phenotype was determined in human CRC cells in which CD133 was co-expressed with B7H1 [82]. All these findings do speculate that CD133+ CCSCs showing EMT phenotype and co-expressing B7H1 may evade immune surveillance. Schmohl et al. demonstrated that innate immune system can be effectively recruited to kill CCSCs using bispecific antibodies targeting CD133. An innovative engineering technology has developed a new anti-cancer molecule. Two fully humanized single chain DNA fragment variable antibodies, recognizing CD16 on NK-cells and CD133 on CSC, were spliced creating a novel drug defined 16 × 133 novel bispecific killer cell engager (BiKE). This molecule simultaneously recognizes antigens to facilitate an immunologic synapse. 16×133 BiKE is a potent engager of the innate immune system capable of inducing NK-cell degranulation and IFN-γ production and mediating selective targeting of CD133+ CSCs. 16×133 BiKE may have therapeutic potential in a clinical NK-cell therapy program for carcinomas, as it could serve as an alternative therapy for drug resistant CSCs by its unique mechanism of action [84].

Analysis of CCSC lysate indicates that they may be a sufficient resource of tumor antigens, and CCSC lysate-based vaccines could stimulate proliferation and differentiation of immune cells overcoming the weak immunogenicity of CCSCs [85,86]. Mice subcutaneously immunized with the CCSC lysate reduced the tumor growth via a target killing of CCSCs. They decrease CD133+ and ALDH+ cells in tumors vs. control vaccine groups, resulting in elevated NK cytotoxicity, perforin production, granzyme B, IFN-γ, memory B cells, and anti-MUC1 antibodies. MUC1 could induce metastasis, cell invasion and proliferation, drug resistance, and angiogenesis in CRC [87]. Further, the antitumor efficacy of CCSC vaccine in MUC1 knockdown was partially impaired [88].

ALDH1+ cells are found at low levels at crypt bases, but increase during progression from normal to APC-mutant adenoma [89]. A high percentage of tumor cells expressing ALDH1 correlate with poor prognosis in various cancers, and display properties of CSCs. ALDH1 is a promising marker to identify CCSCs and a potential candidate for CSC-directed therapy, due to the low expression of ALDH1 in normal colon compared to tumor [90,91].

CD44 is a cell surface glycoprotein involved in cell–cell interactions, adhesion of the cytoskeleton to the extracellular matrix and cell migration [15,92] and, depending on WNT signaling, its overexpression is an early event in the transformation of adenoma to CRC. CD44+/CD24−/CD133− cells form the most aggressive colon tumors, removing the requirement of CD133 in tumor onset [93]. Numerous studies have isolated and characterized EpCAM+/CD44+ cells from CRCs. Dalerba et al. found that normal colon and CRC both contain two populations of cells: EpCAM^High^/CD44+ and EpCAM^Low^/CD44−. Only the first one develops tumors when injected into non-obese diabetic/SCID mice [15]. Using a FACS-based selection, Kemper et al. identified in primary CRCs, a small population of EpCAM+/LGR5+ cells. Spheroid cultures resulting from primary CRC are enriched for CSCs and express high levels of LGR5, while cellular differentiation reduces LGR5 expression. The LGR5^high^ CRC cells are more clonogenic and tumorigenic than LGR5^low^ CRC cells. LGR5 overexpression results in higher clonogenic growth, indicating that LGR5 is a new functional marker for CCSCs [94].

OCT4, SOX2, and NANOG are transcription factors that play a main role in the regulation of pluripotency and implement the self-regulation of their expressions by linking to their promoting regions [95]. OCT4 influences embryogenesis, stem cell maintenance, tumor growth, and metastasis [95,96]. Though it is an important CSC marker, its expression has also been noted in colonic normal tissue. OCT4 expression has been principally identified in the cytoplasm of CRC cells, suggesting a drive factor of recurrence, presumably by preventing apoptosis [96]. OCT4 expression in CRC was present in cells which are undergoing EMT, a key stage in progression and metastasis, and increasing the cell stem-like phenotype [97]. SOX2 is reported to prevent differentiation of neural progenitor cells and to be overexpressed in CRC stage III [96]. Talebi et al. comparing normal colon, dysplastic polyps, and adenocarcinomas identified a significant correlation between SOX2 expression and CRC [98]. High level of SOX2 in CRCs positively correlates with metastases and lymph node infiltration. SOX2 controls OCT4 expression, and this combination of transcription factors promote pluripotency [95]. NANOG also influences pluripotency through transcriptional control [96]. Ibrahim et al. found NANOG in a subpopulation of colon epithelial cells in primary CRC. The OCT4/SOX2 complex regulates NANOG expression and controls the expression of pluripotency-related genes [99]. The high expression of NANOG is correlated with poor prognosis in CRC patients [95].

### Pathways Regulating Intestinal Stem Cell and Cancer Stem Cell Functions

The increasing knowledge of ISC biology has uncovered the molecular mechanisms and possible crosstalk/feedback between signaling pathways that directly influence intestinal homeostasis and tumorigenesis.

Solid evidence indicates that WNT/β-catenin pathway is the most relevant molecular signaling in controlling cell fate through the crypt–villus axis. Mutations in key mediators of the WNT pathway were present in about 90% of CRC [100]. Loss of APC frequently occurs in heritable and sporadic CRC. It transforms epithelial cells through activation of the WNT signaling, leading to the inappropriate stabilization of β-catenin. Rare mutations of the scaffolding protein AXIN2, or β-catenin that remove its N-terminal Ser/Thr destruction motif, underscore the central role of the inappropriate persistence of β-catenin/TCF complex in the epithelial cell transformation [101,102,103,104]. In CRC cells, a constitutive activation of the β-catenin/TCF complex drives a transcriptional program regulating cell fate, cell proliferation, and stem cell maintenance, establishing the primary transforming CRC event. This program is physiologically expressed in crypt stem/progenitor cells. The progenitor proliferating cells at the bottom-crypt accumulate nuclear β-catenin with a consequent hierarchical target genes expression. As the cells reach the mid-crypt, the reduction of β-catenin/TCF activity results in cell cycle arrest and differentiation. The c-MYC downregulation releases p21(CIP1/WAF1) transcription, which sequentially mediates G1 arrest and differentiation. Cells with APC or β-catenin aberrations become independent of the physiological β-catenin/TCF signals. As a consequence, they continue to behave as crypt progenitor cells in the surface epithelium, giving rise to CRC. Thus, the β-catenin/TCF complex represents the switch that controls proliferation/differentiation in healthy and malignant intestinal epithelial cells [105].

Increased WNT activity was identified in the CD133+ CCSC and in LGR5+ ISC, which are supposed to be the origin of adenocarcinomas [106,107]. LGR5 is a WNT pathway target gene and a well-established ISC marker [26]. APC deletion in LGR5+ colonic crypt base SCs induce the rapid transformation into adenomas: Aberrant WNT activity in the LGR5+ SC compartment might be responsible for the tumor-initiating process [107,108]. LGR5 positivity is sufficient for isolation of CSC fraction from tissue defining the CSCs state. CRCs with cells overexpressing LGR5 showed a strong metastatic and recurrence potential [109]. A comparative gene expression analysis between primary CRC and its liver metastases tissues showed ASCL2 overexpression, an ISC marker, and WNT target gene in the metastatic tumors together with several other WNT-induced ISC markers, including LGR5, EphB3, ETS2, and SOX96 [110]. In summary, the canonical WNT signaling is involved in the maintenance and expansion of CSCs. It acts directly on CSCs themselves and indirectly through CSC-stromal/immune interactions. The activation of WNT signaling directly influences the expression of stemness signature genes.

Interactions between WNT signaling and other pathways have been identified and contribute to the early carcinogenesis, CSC maintenance, and CRC progression.

In CRC, c-MET overexpression is usually correlated with tumor progression, metastasis, and poor prognosis [111]: Seemingly, accounted for by enhanced c-MET transcription driven by aberrant WNT signaling. c-MET expression is prevalent at the invasive front of the tumor [112], where the neoplastic cells interact with HGF expressing cancer-associated fibroblasts and macrophages, inducing cell growth and survival, cell motility, and tumor metastasis. c-MET activation might promote WNT signaling inducing stemness on tumor cells [113], by involving AKT-mediated GSK3β activation and leading to phosphorylation of LRP5/6 and stabilization of β-catenin [114,115].

BMP and WNT signaling show multiple interconnections in CRC development. SMAD4 downregulation is correlated with β-catenin overexpression, while functionally the BMP signaling mediates by SMAD4 inhibits β-catenin expression [116]. BMP/SMAD signaling stabilizes the hyperproliferation of ISCs induced by WNT signaling suppressing the expression of SC genes, without a direct interference with WNT signaling [117]. Gene expression analysis of SMAD4, TP53, and β-catenin has found nuclear β-catenin overexpression in the 84% of the CRC showing SMAD4-downregulation and/or TP53 deficient, suggesting that the SMAD4 and TP53 activities control the effect of BMP signaling on the WNT pathway [118]. A regulatory network has been identified between BMP and WNT pathways mediates by PI3K/AKT signaling, which controls SCs behavior and transition of quiescent and activated states, producing a balanced control of SC self-renewal through regulating the β-catenin convergent downstream component [119].

The Hippo pathway was recently included between the signaling involved in CRC pathogenesis and its deregulation associated with poor patients’ prognosis [120]. The crucial interactions between Hippo and WNT signaling maintain cell homeostasis and control ISC proliferation and CRC development. Varelas et al. observed a negative crosstalk between Hippo and WNT signaling by interaction of TAZ and DVL in the cytoplasm, which inhibits CK1 binding and WNT3A-induced DVL phosphorylation, thereby inhibiting the WNT3A-induced transcriptional response. In CRC, the Hippo signaling is repressed, leading to increased nuclear TAZ and reduced TAZ-DVL complex, which results in increased WNT-stimulated DVL phosphorylation, β-catenin nuclear accumulation, and induction of WNT-target genes [121]. Azzolin et al. found that in the absence of WNT signaling, the activation of the β-catenin destruction complex is responsible for TAZ degradation in the cytoplasm [122]. Nuclear YAP physically forms a complex with β-catenin, resulting in the cooperation of YAP/TAZ with β-catenin to induce cell proliferation and promote tumor development.

Aberrant RSPO/LGR signaling is frequently associated with CRC. In a recent model, a crosstalk has been identified between RSPOs protein and WNT signaling that is related to cell surface clearance of the homologous transmembrane E3 ubiquitin ligases RNF43 and ZNRF3, identified as negative modulators of the WNT signaling [123,124]. Without RSPOs, a negative feedback maintains both canonical and non-canonical WNT signaling regulation due to the activity of ZNRF3/RNF43 complex that favors the ubiquitination and degradation of WNT receptors. These ligases decrease the availability of FZD promoting membrane clearance through endocytosis and ubiquitin-mediated degradation. With RSPOs, the assembling of a specific RSPO/LGR/RNF43/ZNRF3 complex suppresses the RNF43/ZNRF3 activity [125] causing the auto-ubiquitination of ZNRF3/RNF43 and improving the availability of FZD receptors [126]. The RSPO/LGR4 complex could improve WNT pathways through the recruitment of IQGAP1 in a ZNRF3/RNF43-independent manner and recruiting the cytoplasmic mediator of WNT signaling DVL. The complex promotes LRP5/6 phosphorylation leading to activation of canonical WNT signaling [127]. Aberrant RSPO/LGR signaling is frequently associated with CRC and a promoter methylation caused RSPO2 downregulation. The RSPO2 anti-oncogenic role on CRC growth depends on the LGR5 presence on neoplastic cells: RSPO2 may function as a tumor suppressor by negatively regulating WNT signaling through a LGR5-dependent manner [128].

Notch signaling maintains the intestinal development and homeostasis regulating differentiation of colonic goblet cells and stem/progenitor cells [129]. In CRC cells, a direct interaction between β-catenin and Notch signaling regulated by Jagged1 expression has been identified. The interaction of β-catenin with Notch1 decreases its ubiquitination, causing an increase of HES-1 gene expression, which is associated with tumorigenesis. The GSK3β is an important junction between WNT and Notch signaling crosstalk, and mediates the phosphorylation of Notch intracellular domain-1 (NICD-1), which localizes it in the nucleus, enhancing its transcriptional activity and stabilizing it [130]. The non-canonical WNT/Ca2+ pathway relates with Notch signaling: Activation of CaMKII by WNT5a promotes phosphorylation of RBP-J-interacting corepressor SMRT increasing promoter activity of Notch-responsive genes [131].

Hedgehog signaling plays a crucial role in CRC pathogenesis. Different members of this pathway such as SHH, PTCH1, SMO, and Gli are shown to be overexpressed in CRC and positively correlated with tumor progression [132,133]. Hedgehog signaling maintains CSCs survival and expansion [134]. A crosstalk between Hedgehog and WNT pathways is important for CRC recurrence and metastatic potential [135,136]: The kinases GSK3β and CK1 worked to phosphorylate and degrade both β-catenin and full-length Gli-3. The latter is recognized by b-TrCP leading to the degradation of C-terminal peptides to generate Gli3R, which subsequently blocks Gli-1 activity [137]. SFRP-1 is a negative regulator of both WNT and Hedgehog pathways, it interacts with Gli-1, but also binds to β-catenin to control their nuclear–cytoplasmic distributions [138,139]. Moreover, both pathways are regulated by common modulators such as TP53, PTEN, SMO, and KRAS [137,140,141].

Single gene mutation of members of WNT and RAS/MEK/ERK pathways often does not induce significant neoplastic transformation while multiple variations of both pathway genes work synergically to promote onset and progression of tumorigenesis. A growing body of evidence indicates that APC and KRAS gene mutations, playing a role in WNT and RAS/MEK/ERK pathways, respectively, cooperatively interact, inducing cell proliferation, neoplastic transformation, CSCs activation, and metastasis. A further mechanism based on the stabilization of β-catenin and mutation status of KRAS was described depending on the status of WNT signaling. The APC loss could explain a positive cooperation due to usage of a common kinase GSK3β in the phosphorylation of β-catenin and KRAS, increasing nuclear β-catenin and RAS/MEK/ERK signaling activation, which induce transcription of target genes as CSCs markers CD44, CD133, and LGR5 and EGFR involved in the progression of malignancy phenotype [142,143].

PI3K/AKT/mTOR signaling is hyperfunctioning in CRC and contributes to keep cell growth and metastasis in CRC [144]. The inhibition of this pathway might represent a potential targeted therapeutic approach [145]. mTOR signals, activated through the RAS/MEK/ERK and PI3K/AKT pathways induced by tyrosine kinase receptors EGFR and IGFR, regulate cell growth and proliferation, cell metabolism by mediating multiple signals: growth factors, nutrients, hormones, and energy/stress status [146,147]. A crosstalk between RAS/MEK/ERK and PI3K/AKT pathways in CSCs has been reported in CRC [143,148]. PI3K/AKT pathway could deactivate GSK3β increasing nuclear β-catenin [149]. The combined expression of CSC markers (CD166, CD44), active mTOR signals (pS6), and WNT signal markers (β-catenin) increase CRC aggressiveness such as metastasis and resistance to chemotherapy via synergic interactions. Specific patterns of CSC markers and β-catenin/mTOR signaling could predict poor survival in stage II CRC [150].

In intestinal epithelium, EphB receptors and ligands are WNT signaling target genes that control cell compartmentalization along the crypt axis, defining crypt–villus boundaries and positioning Paneth cells at the bottom-crypt [50]. EphB receptors are suppressors of CRC progression, showing that the entity of EphB2 silencing correlates inversely with patient survival [151,152]. The EphB downregulation in CRC, in spite of the constitutive activation of the WNT signaling, should suggest a mechanism of transcriptional silencing of EphB genes that acts in a dominant way over the β-catenin/TCF complex.

Inflammation-related signal is a crucial regulator in cancers driven by WNT signaling [153]. Gavert et al. demonstrated that the neural cell-adhesion molecule L1CAM, which is a target of WNT signaling, is exclusively expressed in the invasive front in CRC, increases cell growth and motility, and promotes liver metastasis through the NF-κB signaling activation. Overexpression of the NF-κB p65-subunit was demonstrated to increase CRC cell proliferation, motility, and metastasis. Using the inhibitor of κB super repressor (IκB-SR), the L1CAM-mediated metastatic process could be blocked [154,155]. The metastatic capacity of CRC relates to a complex between the cytoskeletal crosslinking protein Ezrin, L1CAM, and the active phosphorylated p65-subunit in the juxtamembrane region of neoplastic cells, suggesting that L1CAM-mediated activation of NF-κB signaling involving Ezrin is a key mechanism of CRC progression [156].

Figure 2 shows a graphical representation of the main complex positive and negative feedback loops existing between WNT signaling and other signaling pathways involved in CRC pathogenesis. We underline their essential role in deregulation of important biological cell processes such as growth and proliferation, apoptosis, invasion and motility, CSCs maintenance, and EMT leading to progression, invasiveness, and therapy resistance.

## 4. MicroRNAs Dysregulation on Colorectal Cancer Stem Cells: An Overview

MiRNAs are a family of small endogenous single-stranded (21–25-nucleotide) noncoding RNAs that regulate gene expression by different feedback mechanisms, such as recognition of proprietary sequences, seed region, 3′UTR or 5′UTR, or open reading frame (ORF) of the mRNA target, triggering mRNA degradation or translational repression [157,158]. One miRNA can target up to several hundred mRNAs, alternatively, the expression of a gene can be controlled by several miRNAs. Deregulation of miRNA expression could affect a multitude of transcripts and influence several functions of the cancer pathway. MiRNAs regulate the expression of gene involved in biological processes whose dysregulation is strongly related to initiation and progression of cancer [159].

The recognition of miRNAs as regulators of gene expression has made them the most widely investigated epigenetic elements involved in cancer: They potentially represent therapeutic targets, but also diagnostic and prognostic biomarkers for malignancy. MiRNA expression level variations were identified between colorectal neoplastic vs. normal tissues. Evidence from in vitro and animal models (nude mice) has shown that anti-tumor “miRNA mimics” inhibit cell proliferation, migration, and induce apoptosis in CRC [160,161] while several miRNAs are indicative of response to chemotherapy [162]. MiRNAs could have critical clinical significance in the diagnosis, treatment, and prediction of outcomes in CRC patients.

MiRNAs are involved in controlling the SCs growth and regulatory mechanisms, such as maintenance, reprogramming, pluripotency, and differentiation by targeting various signaling pathways [163,164]. MiRNAs expression profiles were related to SCs vs. differentiated cells tissue [165]. MiRNAs expression variations causing deregulation of signaling pathways in CCSCs, influence CSC features, EMT, angiogenesis, metastasis, and pharmaco-resistance processes [166]. MiRNAs regulate functions of CCSCs interacting with the main signaling pathways required for maintenance of SC pluripotency (WNT/β-catenin, Notch, BMP, inflammatory signaling pathways, etc.) [167].

MiR-21 was one of the most deeply studied oncomiR in CCSCs. MiR-21 is overexpressed in chemotherapy-resistant (CR) CRC cells highly enriched in CSCs. It controls tumor growth, recurrence, invasion, and metastasis acting on downregulation of oncosuppressor genes including PTEN, which is normally involved in the rule of SCs self-renewal. A strong downregulation of PTEN in SCID mice xenografts of miR-21-overexpressing HCT116 cells leads to an increase of 60-80% in the self-renewal capacity of CCSCs. Colonospheres highly enriched in CCSCs show miR-21 overexpression and PTEN downregulation [168], and their downstream effectors, such as TIAM1, promote EMT and cell migration [169]. Conversely, miR-21 downregulation results in a significant reduction in CD44 expression and regulation of the tumorigenic properties of colonospheres enriched in CCSCs [168]. Interesting results were obtained using the difluorinated curcumin (CDF), which inhibits the growth of 5-Flurouracil-Oxaliplatin-resistant HCT116, HT-29, and metastatic SW620 CRC cell lines, decreasing miR-21 levels and restoring PTEN levels with following reduction in AKT phosphorylation miR-21/AKT/PTEN axis could be responsible of differentiation of CRC cells and enhance susceptibility to therapeutic regimens [168]. MiR-21 plays an important regulator role in CRC stemness cells by activating WNT/β-catenin signaling modulated by downregulation of TGFβR2, which is a direct target of miR-21 [170]. Recently, a negative feedback between miR-21 and miR-145 mediated by the RAS/MEK/ERK pathway has been identified and related to CSC proliferation and/or differentiation in CRC cell lines. It has been hypothesized that miR-21 increases RAS/MEK/ERK signaling activity leading through RREB1 to the repression of miR-143/145 cluster, and that miR-145 inhibits miR-21 transcription through blocking KRAS and decreasing AP1, which is the main transcription factor of miR-21. MiR-21 controls its upregulation by a positive feedback loop with AP1. In vivo experimentation proved that growth inhibition reduces CSCs proliferation and differentiation, as evidenced by CD44 downregulation and CK20 induction following miR-145 overexpression or miR-21 downregulation [171]. MiR-21 overexpression was observed in CRC neoplastic tissues, serum, and stool, and increased with disease progression and between TNM stages of CRC [172,173,174]. MiR-21 overexpression in serum and tissue has been significantly related with poor response to chemotherapy and disease-free survival [175]. Notably, miR-21 was downregulated in serum from patients who underwent curative surgery [174]. Bullock et al. demonstrated that deregulation of miR-21 is a stromal phenomenon in CRC, highlighting a potential mechanism through which upregulated stromal miR-21 increases invasion in vitro. MiR-21 overexpression in stromal fibroblasts induces TGF-β-dependent fibroblast-to-myofibroblast trans-differentiation and negatively controls RECK protein expression, which regulate negatively MMP2 via post-transcriptional mechanisms, then enhanced tumor invasion through upregulated MMP2 activity [176]. MiR-21 is also implicated in drug interaction with Regorafenib, a broad-spectrum kinase inhibitor approved for CRC treatment in combination with classical chemotherapeutic drugs. An in silico and in vitro integrated analysis exhibited a direct intermolecular interaction between Regorafenib and miR-21 pre-element to stabilize miR-21 pre-element and prevent RNase Dicer-mediated cleavage of the pre-element to mature miR-21. Describing a potential alternative mechanism for anti-CRC treatment with Regorafenib [177].

MiR-221 is a CCSC-related miRNA, overexpressed in 90% of CRC and positively associated with advanced TNM stage, local invasion, and poor prognosis [178,179]. MiR-221 overexpression in plasma and stool could be considered as a non-invasive diagnostic and prognostic marker correlated with poor overall survival in CRC patients [180,181]. MiR-221 binds to the 3′-UTR of the CDKN1C/p57 mRNA in an incomplete complementary pairing pattern and inhibits mRNA translation without a complete degradation. In vitro experiments confirmed that miR-221 overexpression promotes the CRC cells growth increasing the S-phase population by CDKN1C/p57 downregulation. MiR-221-specific inhibitor inhibits cell proliferation and induces apoptosis in CRC, inducing overexpression of CDKN1C/p57 [178]. A miR-221/222-mediated positive feedback loop provides constitutive activation of NF-κB and STAT3 pathways in CRC, suggesting a link between inflammatory signaling and oncogenic addiction. MiR-221/222 control NF-κB and STAT3 signaling by binding to and stabilizing RELA mRNA. MiR-221/222 upregulate both RELA and STAT3 protein binding the PDLIM2, reducing the ubiquitination and degradation of both proteins [182]. MiR-221 overexpression improves invasion and metastasis in CRC through targeting RECK [183], and its inhibition enhances the radiosensitivity of CRC cells inducing PTEN overexpression [184]. A mechanistic link between miR-221 and QKI highlighted their key role in regulating CSC properties in CRC. MiR-221 is mainly expressed in the EpCAM+/CD44+ population and represents a required component of the tumor growth in vivo. QKI-5 has been identified as a functional target of miR-221 and a tumor growth suppressor in vivo. These results suggest that the functional interaction between miR-221 and QKI represents a molecular network involved in the regulations of CSC biology in CRCs [185].

Monzo et al. found that the miR-17-92 cluster and its target gene E2F1 exhibit a similar expression pattern comparing human colon tissue and CRC regulating cell proliferation in both tissues. In situ hybridization identified miR-17-5p overexpression in the crypt progenitor compartment, suggesting a controller role in colon stem/progenitor cell proliferation and differentiation [186]. MiR-17-mediated CYP7B1 regulation promotes EMT and the formation of a stem cell-like population in human CRC cells [187]. Expression profile in exfoliated colonocytes isolated from feces in CRC and healthy patients, showed miR-17-92 cluster and miR-135 overexpression in CRC patients vs. healthy volunteers. MiRNA expression profile of the isolated colonocytes may be useful for CRC screening [188]. MiR-18a upregulation in in vitro models and tissues/sera of CRC patients, revealed association to CSC phenotype, metastasis and age, suggesting its role as metastatic biomarkers in CRC [189]. CSC-related miR-17 expression increases during progression from the primary CRC to liver metastasis [190]. MR-92a overexpression is present in chemoresistant CRC tissues and cells. In vitro experimentation showed that the miR-92a expression conferred resistance to 5-Fluorouracil (5-FU), while antagomiR-92a significantly enhanced the drug sensitivity of CRC cells in vivo, establishing a role of miR-92a in supporting the development of chemoresistance in CRC. A newly IL-6/STAT3/miR-92a/WNT axis is involved with chemoresistance, where IL-6/STAT3 pathway raise up miR-92a expression by direct target of its promoter, switching on WNT/β-catenin signaling activation by targeting KLF4, GSK3β, and DKK3, and subsequently stimulating the stem-like phenotype of CRC cells [191]. Viswanathan et al. demonstrated that miR-92a downregulates LRIG1, that there exists a positive correlation between miR-92a levels and CRC cell numbers, and that miR-92a overexpression was identified in ALDH-positive CCSCs, which do not express LRIG1. Considering that LRIG1 is known as a pan-ERBB negative regulator and that LRIG1 promotes SC quiescence especially in intestinal system, it has been hypothesized that miR-92a overexpression, by LRIG1 downregulation, decreases differentiation and increases proliferative potential of ALDH-positive CSCs during CRC development [192]. MiR-492, miR-200a, miR-338, miR-29c, miR-101, miR-148a, miR-92a, miR-424, and miR-210 were proven to be promising diagnostic and prognostic markers in the clinic outcome of CRC patients [193].

MiR-371/373 cluster is a specific regulator of SCs maintenance, and strongly dysregulated in various cancers, particularly in CRC. WNT/β-catenin signaling transactivated miR-372/373 in a β-catenin/LEF1/miR-372-373/DKK1 regulatory feedback loop way suggesting that miR-372/373 upregulation provides to CSC-like properties in CRC [194]. MiR-371/373 overexpression improves the stemness of CRC cells by enriching the CD26/CD24-positive cell population responsible for chemoresistance, self-renewal, and metastases. A mechanistic model identified as responsible for miRNA-372/373-induced stemness includes upregulation of NANOG and Hedgehog stemness-pathways and repression of NF-kB, MAPK/ERK, and VDR differentiation-related pathways. Important regulator genes of cell differentiation such as SPOP, VDR, and SETD7 are target genes of miR-372/373, and their downregulation reveals poor differentiation of CRC cells. MiR-372/373 improves CRC cell stemness by repressing differentiation genes [195]. Contrariwise, miR-371-373/TGFβR2/ID1 signaling axis represents an innovative regulatory mechanism of self-renewal and metastatic colonization in TICs obtained by primary cultures enriched in CCSCs. The loss of miR-371/373 cluster expression and the consequent induction of TGFβR2 and ID1 signaling improve the self-renewal capacity and metastatic outgrowth potential of disseminated TICs [196].

The bone marrow-derived mesenchymal stem/stromal cells (BM-MSCs) play important roles in tumor development through the release of cytokines or exosomes. BM-MSC-derived exosomes contained distinct miRNAs including miR-142-3p, which promotes the CSCs phenotype in CRC by targeting Numb and thus promoting the Notch pathway activation [197]. The treatment with Antrodia cinnamomea (AC) decreases the number, cell viability, inhibits self-renewal capacity, and induces apoptosis of colon tumor spheres, due to the downregulation of cancer-associated stemness marker such as β-catenin, SOX2, NANOG, and of EMT-related genes such as vimentin and MMP3, and upregulation of miR142-3p. AC combined with 5-FU increase the sensitivity of CRC cells to chemotherapeutic drugs and synergistically suppress the neoplastic cell viability by stimulating the expression of apoptosis-related genes and suppressing inflammation and metastasis-related genes through miR142-3p [198].

MiR-196b-5p was markedly upregulated in CRC tissues and in the serum exosomes of CRC patients and its high expression correlates with poor survival. MiR-196b-5p stimulates stemness and 5-FU-chemoresistance of CRC cells, giving rise to activation of STAT3 signaling by targeting negative regulators of the pathway such as SOCS1 and SOCS3 [199].

MiR-26b is associated with high lymph node metastasis in CRC patients. It promotes metastasis and invasion in the CRC cell lines contributing to induce EMT and stem cell-like phenotype in a subpopulation of CRC cells by simultaneously suppressing PTEN and WNT5A mRNAs [200]. Preclinical CRC model and retrospective studies have identified new potential prognostic and predictive targets, such as EphA2/EFNA1/EGFR/PTPN12/ATF2/mir-200a/mir-26b, which could be helpful in selecting CRC patients with poor prognosis and Cetuximab (CTX)-resistance [201].

First evidence of miR-34a as a tumor suppressor in cancer dates back to 2008. A positive feedback loop was identified in which TP53 induces miR-34a expression, which through inhibition of SIRT1, increases TP53 activity in CRC cell lines [202]. The Notch pathway is affected by miR-34a expression: This signaling shows a major role in cell fate determination either during development or oncogenesis. MiR-34a binds the Notch receptors mRNA, reducing protein levels and dampening downstream Notch signaling. The Notch signaling is a critical regulator of asymmetric/symmetric division in several normal SCs, including colon SCs. The decision of a CCSC to perform either symmetric or asymmetric division is strongly depended on miR-34a expression levels, which acts as a bimodal switch to target Notch in CCSCs, regulating the choice of daughter cells to self-renew or to differentiate during division [203]. Furthermore, the activation of a conditional miR-34a allele in DLD-1 CRC cells reduces c-KIT expression and suppresses sphere formation, demonstrating that miR-34 directly targets c-KIT, interfering with c-KIT-mediated effects on CRC cells as well as expression of markers of stemness and β-catenin, activation of ERK signaling, invasion and metastasis and chemoresistance [204]. A further feed-forward regulatory loop between SNAIL/miR-34a/ZNF281 which regulates the EMT process was identified. ZNF281 interacts with NANOG, OCT4, SOX2 transcription factors, and c-MYC regulating pluripotency and stemness [205]. A new regulatory model conducted by Lnc34a, strongly enriched in CCSCs, causes miR-34a spatial imbalance by directly targeting it and promoting asymmetric division of CCSCs. Lnc34a silences miR-34a gene, independently of TP53, by recruiting DNMT3a via PHB2 and HDAC1 to methylate and deacetylate the miR-34a promoter. Besides, Lnc34a levels affect CCSC self-renewal and growth in xenograft CRC models [206]. More importantly, miR-34a regulates chemosensitivity in CRC [10]. CRC patients treated with Oxaliplatin (OXA)-based chemotherapy showed miR-34a downregulation and TGF-β and SMAD4 overexpression, as well as in vitro experiments in OXA-resistant or not CRC cells by OXA treatment. OXA-induced miR-34a downregulation increased drug resistance activating macro-autophagy in CRC cells. Expression of SMAD4 and miR-34a in CRCs shows a significant inverse correlation, while overexpression of miR-34a inhibits macro-autophagy activation by targeting SMAD4 through the TGF-β/SMAD4 pathway [207]. Regorafenib treatment reduces stemness phenotypes, as well as number of colonies, tumor spheres, and stem-like phenotypes and WNT1, mTOR/STAT3, Notch1 markers. Mechanistically, regorafenib-treated CRC cells showed overexpression of tumor suppressor miRNAs, especially miR-34a, suggesting that regorafenib treatment suppresses CCSCs via the induction of miR-34a levels, explaining why patients with metastatic and drug-resistant responded towards the treatment of regorafenib [208]. MiR-34a in combination with paclitaxel strongly decrease cells viability in CRC cell lines, through targeting and downregulating SIRT1 and BCL2 genes [209].

The ZEB1-2 transcription factors are critical EMT activators, unlike to the miR-200 family (miR-200a, miR-200b, miR-200c, miR-141, and miR-429), which induces epithelial differentiation. A reciprocal feedback loop between the members of the TGF-β/ZEBs/miR-200f axis has been identified. EMT prevalently controls cellular motility, but it is also related to SC properties, and miR-200f plays an important role as an SC regulator through negative EMT modulation [210]. MiR-200f is one of the most remarkable tumor suppressor miRNA in various types of cancer [211]. In CRC patients, miR-200a downregulation represents an independent prognostic factor, based on differences in the EMT and SC formation capacity [212]. High miR-200c serum levels might represent an interesting noninvasive prognostic CRC marker [213]. High levels of miR-141 in plasma are associated with the capacity to detect stage IV CRC when combined with CEA marker, and were negatively associated with overall survival in patients [214]. In CRC cells, the miR-200c downregulation increases the expression of SCs markers, such as CD166 and CD133, and improves SCs characters. MiR-200c and SOX2 reciprocally control their expression through a feedback loop: MiR-200c suppresses the expression of SOX2 to block the activity of the PI3K/AKT pathway [215]. The EMT inducer ZEB1 could trigger EMT and preserve stemness, enhancing the invasiveness of cancer cells in the whole body, and promoting stemness-associated growth and differentiation in metastasis.

Mechanistically, ZEB1 connects EMT-activation and stemness-maintenance by suppressing stemness-inhibiting miRNAs, such as miR-200c, miR-203, and miR-183 that cooperate to suppress expression of SC factors in neoplastic cells. It can be concluded that ZEB1 is a promoter of CCSs migration [216]. ZEB1 overexpression is induced by SPRY2 by induction of ETS1 transcription factor and repression of miR-200 family and miR-150 [217]. ASCL2 is a downstream target of WNT signaling that controls the fate of intestinal cryptic SCs and CRC progenitor cells. Its downregulation promotes the reversion of mesenchymal phenotypes by increasing the miR-200f levels through a direct binding to miR-200b-a-429 promoter [218]. NANOG is a SC gene that could potentiate tumorigenic cooperating with other molecules. It demonstrated its direct repression on transcription of the miR-200c and miR-200b genes, which mediated NANOG-induced EMT [219]. miR-200c is also involved in chemotherapeutic drug resistance. The miR-200c repression in CRC leads to an increase in 5-FU chemoresistance and early EMT while decreasing the levels of E-cadherin and PTEN protein [220]. The inhibitory effect of Zerumbone on the EMT process and CSC markers, in the presence or absence of miR-200c, could be a promising candidate for reducing the risk of CRC progression by suppressing the WNT/β-catenin signaling via miR-200 [221].

Low miR-302a expression especially in CTX-resistant cells, patient-derived xenografts, and CRC tissues, correlates with poor overall survival in CRC patients. Restoring miR-302a expression in vitro and in vivo inhibits metastasis by directly targeting NFIB, which is a transcriptional activator of ITGA6, and restores CTX-sensitivity directly suppressing CD44, which induces CSCs properties and activates EGFR-dependent MAPK and AKT signaling [222]. MiR-302a overexpression is a useful strategy for inhibiting IGF1-R, inactivating AKT, and enhancing 5-FU-induced cell death and viability inhibition in CRC cells [223].

MiR-20b is correlated with clinicopathological features and survival rate in CRC. It reduces 5-FU-resistance by suppressing the ADAM9/EGFR signaling in CRC cells [224]. The expression of miR-20b-5p is negatively correlated with MALAT1 and OCT4 levels in CRC cells, assuming that there might be some targeted regulatory axis between them. MALAT1, a competing endogenous RNA, might interfere with miR-20b to mediate the expression of OCT4, a transcription and maintenance factor for stemness, growth, and tumor metabolism [225].

MiR-451 controls the ability of CCSCs to self-renew and resist drugs. Restoring of miR-451 results in reduction of colon sphere formation and growth and inhibition of tumorigenicity in vivo by inducing downregulation of COX-2 and WNT pathway, while the sensitization to the irinotecan in CSCs is induced by the targeting of ATP-binding cassette [226]. MiR-451 overexpression reduces CRC cell proliferation and accumulation at the G0/G1 phase of the cell cycle and increases apoptosis, with a corresponding cellular arrest in the G2/M phase. MiR-451 reduces the expression of OCT4, SOX2, and SNAIL, indicating its role in SCs and EMT regulation [227].

MiR-137 inhibits stem function in various types of SCs, such as neural SCs and embryonic SCs [228]. MiR-137 directly regulates MSI1 and is inversely correlated in a panel of CRC cell lines and rectal tumors vs. paired normal mucosa. In vitro, miR-137 over-expression reduces MSI1 expression, cell and tumor sphere growth, and colony formation while restoring miR-137 expression in xenograft tumor models minimizes tumor growth in vivo [229]. MiR-137 suppresses CCSCs tumorigenicity, and its expression in normal SCs suppresses uncontrolled cell proliferation through inhibition of DCLK1 expression [230].

Initial findings identified low miR-93 expression levels in human CCSCs vs. differentiated CRC cells, indicating the involvement of miR-93 in the development or replication of CCSCs. Restoring miR-93 inhibits cell proliferation and colony formation of CCSCs [231]. Mir-93 downregulation represents a prognostic factor for CRC: It is significantly associated with adverse clinicopathologic features and short overall survival in CRC patients [232]. Recent data identify a lncRNA–miRNA functional network on a subpopulation of highly tumorigenic human CRC cells, in which miR-93 directly binding the lncRNA LOCCS reducing cell proliferation, invasion, migration, and generation of tumor xenografts [233].

MiR-215 is involved in the transcriptional network regulated by CDX1 and facilitates the repression of cell cycle and stemness genes downstream of CDX1-mediated differentiation switch. MiR-215 targets BMI1 mRNA, which varies inversely with CDX1, promoting stemness and self-renewal [234]. MiR-194/miR-215 cluster suppresses growth and attenuates the stemness of intestinal tumor organoids [235]. The lncRNAs UICLM acts as endogenous molecular sponges to compete with miR-215 to regulate ZEB2 expression involved in EMT phenotype and stem cell properties and affecting cell invasion and liver metastasis in CRC cells [236]. MiR-215–5p is one of the main hypoxia-induced miRNAs in different primary colon TICs cultures. It showed strong tumor- and TIC-suppressor potential, acting as a negative feedback regulator of hypoxia-induced TIC activity. LGR5 has been identified as a new hypoxia/miR-215 downstream target: Low expression could suggest a selective depletion of tumorigenic cells in specific cultures [237].

In CRC, miR-203 downregulation was correlated with advanced TNM stage and lymph node metastasis. CRC patients with miR-203 downregulation had worse prognosis than patients with a high miR-203 expression. MiR-203 overexpression abolishes in vitro cell proliferation and invasion, promotes apoptosis, and inhibits in vivo tumor growth and metastasis, by directly blocking EIF5A2 expression [238]. The overexpression of miR-203 enhanced the sensitivity of paclitaxel to TP53-mutated CRC cells, through the inhibition of AKT2 activity. MiR-203 represents an inhibitor not only for tumor growth and metastasis, but also revers chemoresistance in CRC [239].

The hyaluronan (HA) is a component of the extracellular matrix of mammal cells that interacts with its receptor to activate c-SRC-dependent signaling that promotes cell migration and tumor progression. CD44+ CSCs obtained by CRC cell lines treated with HA show downregulation of miR-203 and overexpression of SNAIL protein and its nuclear accumulation. Direct binding between SNAIL and the E-box-containing region of the miR-203 promoter is responsible of CSCs maintenance in CRC [240].

The role of miR-16-5p was evaluated in in vitro experiments revealing that the BMSCs-derived exosomes overexpressing miR-16-5p inhibit proliferation, migration, and invasion, and simultaneously stimulate the apoptosis of the CRC cells via downregulation of ITGA2. BMSCs-derived exosomes overexpressing miR-16-5p repressed the tumor growth [241]. MiR-16 targets survivin at both mRNA and gene levels. A regulatory loop between miR-16 and TP53 has been identified: TP53 inhibits the cyclin D1, CDK6, and survivin expression through upregulating miR-16 in CRC cells. This process blocks proliferation and induction of apoptosis of CRC cells through the intrinsic apoptosis pathway [242,243].

Shen et al. reported that the miR-139-5p levels decrease with the increase of clinical malignancy features in CRC patients [244]. They confirmed its regulatory role in CRC cell invasion and metastasis and in the reversal of multidrug resistance in CD133+/CD44+ CCSCs. A multiple drug resistance (MDR) cell model, which overexpresses CD44 and CD133, demonstrated miR-139-5p downregulation and Notch1 overexpression. The latter is an important protein for SC maintenance and function, as a direct target of miR-139-5p, both in vitro and in a knockout mouse model, maintains cell drug resistance [245]. MiR-139-5p targets the WNT/β-catenin/TCF7L2 downstream effector E2-2 in CCSCs, regulating the CCSCs stemness maintenance and metastasis, especially by EMT [246].

Some evidence underlies a close link between the oncosuppressor miR-147 and EMT, which is reverted by the ectopic expression of miR-147 in CRC cells. A miR-147/TGF-β signaling network is a regulator of EMT and progression of CRC. High levels of miR-147 in CRC cells decreases AKT phosphorylation inducing EMT reversion, cell cycle arrest, and recovering the epithelial EGFR inhibitor sensitivity at gefitinib in CRC resistance cells [247]. The ectopic expression of miR-147 in CRC cell lines, inhibits stem cell-like traits with downregulation of CCSC markers, OCT4, SOX, and NANOG. Mechanistically, miR-147 inhibited stem cell-like traits in CRC cells by suppressing EMT via the WNT/β-catenin pathway [248]. Recently, the inhibitory effect of miR147 overexpression on CRC cells growth has been related to RAP2B gene levels that represent a target for miR-147 [249].

Strong and complex regulatory networks have been identified between dysregulated miRNAs and disruption of the cell signaling balance involved in CSCs survival and proliferation. An increasing number of studies have established that miRNAs, acting as tumor suppressor or oncogene, could moderate and/or deregulate various signaling pathways involved in regulatory mechanisms of CSCs as well as cell cycle, self-renewal, and differentiation, EMT and drug resistance (Figure 3).

The miRNAome deregulation is responsible of abnormalities in gene expression and signaling pathways functions in CRC, shows differentiated miRNA expression profiles associated with different features of CRC and represents the mechanism to induce CRC initiation and progression. Table 1 provides an illustration of oncomiR and anti-oncomiR involved in CRC pathogenesis.

## 5. Conclusions

A growing body of evidence suggests that the heterogeneous nature of CRC is related to CSCs, neoplastic cells with stemness behaviors responsible for tumor progression, recurrence, and resistance to therapy. It is important to establish the critical role of miRNAs as modulators of the stemness features of CCSCs, achieved through modulation/deregulation of the key SC signaling pathways. Here we discuss the diagnostic/prognostic clinical utility of several miRNAs, the regulation of miRNA-mediated chemosensitivity in CCSCs highlighting the potential predictive role of miRNAs in CRCs. Conventional chemotherapy typically relies on the elimination of differentiated CRC cells while failing to eliminate the CCSCs with tumor persistence. A treatment approach combining targeted drugs with conventional chemotherapy has achieved more favorable results. Interestingly, CSCs have specific surface markers that should be targeted in development of therapies, in addition to targeting members of the signaling pathways, the tumor microenvironment, modifying the epigenetics of CSCs, and focusing on eliminating CSCs by stimulating the immune system. Stem cell miRNAs, miRNA-mimics, and miRNA-antagonists could be potential targets for development of individualized therapies and considering for clinical application. Considering that drug-targeted renewal and differentiation processes of CSCs are insufficient, stem cell miRNA-targeted therapy should provide advantages towards development of new therapeutic options against CRC. Moreover, targeting of CCSCs should increase chemosensitivity and reduce metastatic potential of neoplastic cells, leading to a better outcome for CRC patients. To confirm the results obtained in CRC preclinical models, clinical translated analysis will be required before a miRNA-based therapeutic approach can be applied into clinical applications.

## Figures and Tables

**Figure 1 ijms-22-01603-f001:**
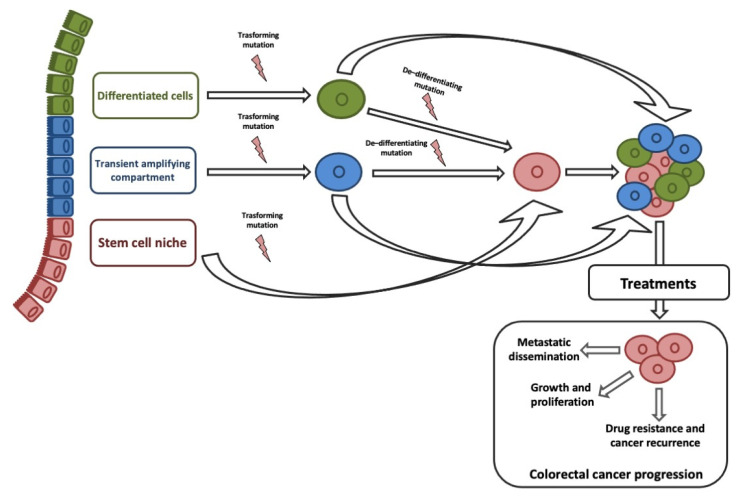
Origin of colorectal cancer stem cells (CCSC) and their role in tumor progression. Colorectal cancer (CRC) cells may originate from intestinal stem cells (ISCs), cells in transit, and terminally differentiated cells gaining a transforming mutation. A single mutation in an ISC could originate a CCSC; at least one transforming, and one de-differentiating mutation would be needed to transform a transit-amplifying (TA) or differentiated cell into a CCSC.

**Figure 2 ijms-22-01603-f002:**
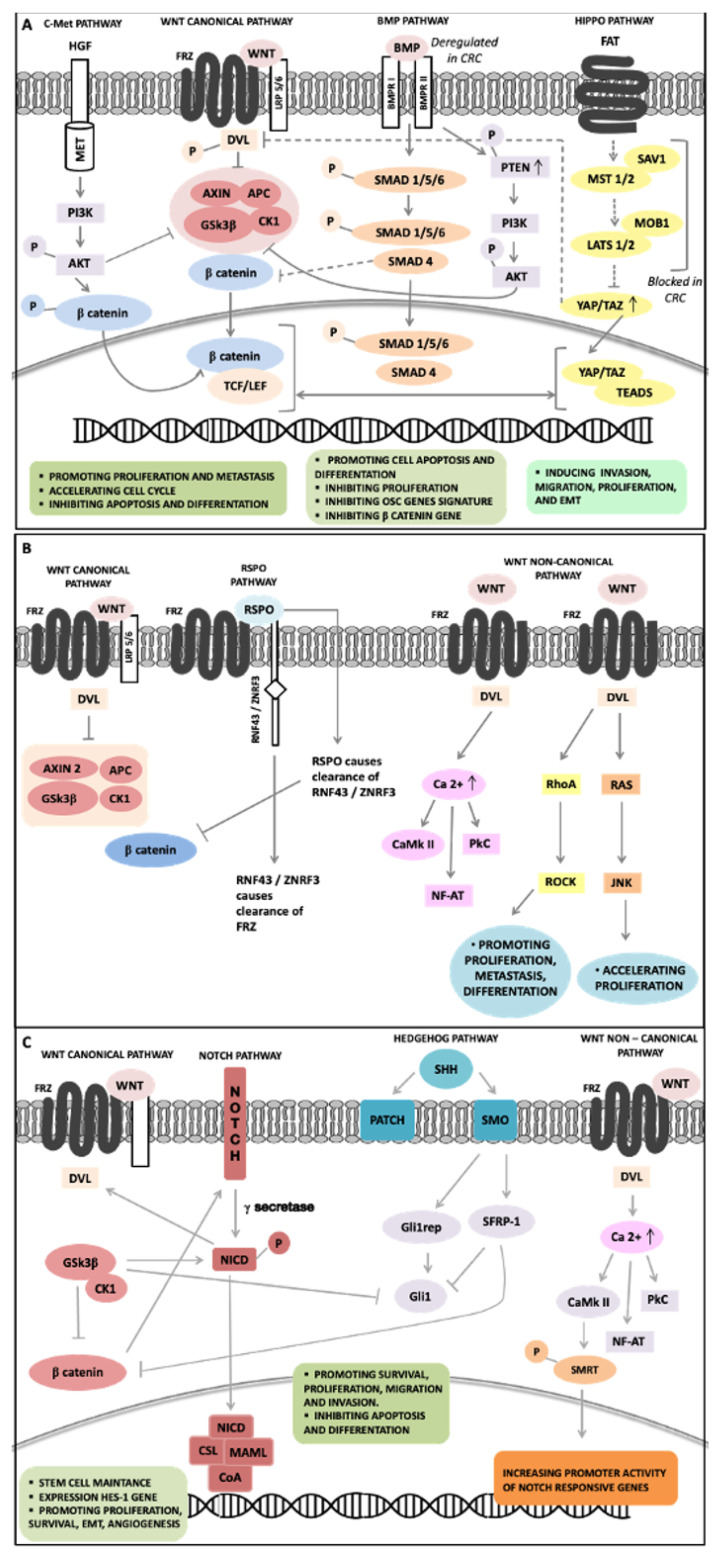
Overview of feedback loops between signaling pathways involved in CRC pathogenesis and controlling stem cell behavior through regulation of β-catenin, a key downstream convergent component. (**A**) c-MET activation induces AKT and GSK3β phosphorylation leading to LRP5/6 phosphorylation and β-catenin stabilization; deregulation of BMP signaling stabilizes β-catenin by downregulation of SMAD4 and PTEN phosphorylation; Hippo signaling repression stabilizes β-catenin reducing the TAZ-DVL complex in the cytoplasm, while the increasing of TAZ/YAP complex in the nucleus cooperates with β-catenin. (**B**) RSPOs induce RNF43/ZNRF3 ubiquitination improving WNT canonical and non-canonical signaling. (**C**) The interactions between Notch and WNT signaling improve Notch activity, GSK3β phosphorylates NICD-1 favoring localization in the nucleus, CaMKII induces phosphorylation of SMRT increasing promoter activity of Notch-responsive genes; the crosstalk between Hedgehog and WNT signaling includes Gli-3 phosphorylation by GSK3β/CK1 blocking Gli-1 activity, SFRP-1 interacts with Gli-1 and β-catenin to control their nuclear–cytoplasmic distributions.

**Figure 3 ijms-22-01603-f003:**
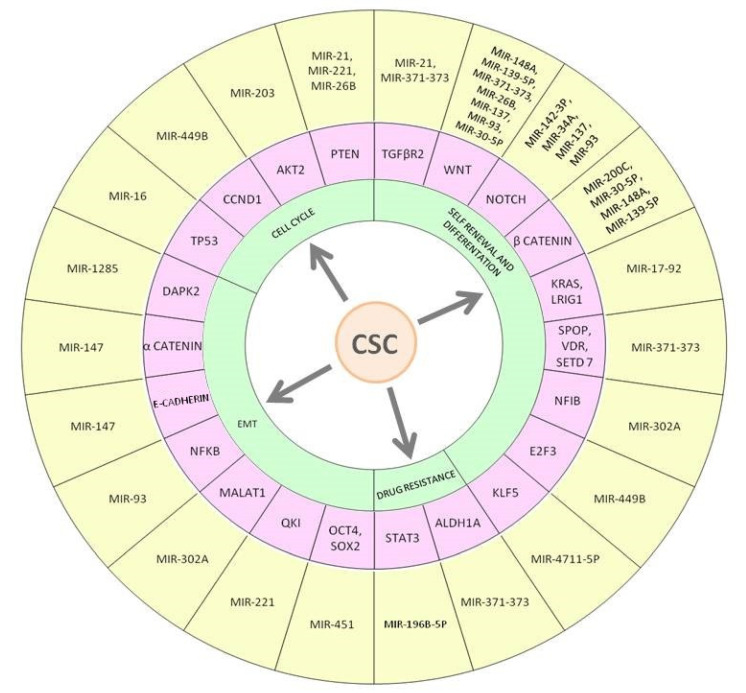
Main miRNAs regulating colorectal cancer stem cell features. The figure shows miRNAs-mRNA interactions that may reduce or promote the development and features of CCSC, such as cell cycle, self-renewal and differentiation, EMT process, and drug resistance linked to the initiation and progression of CRC. MiR-21 is an oncomiR that promotes CCSCs development by interacting with PTEN and TGFβR2. A regulatory loop has been identified between miR-16 and TP53, its deregulation in CCR favors development of CCSCs.

**Table 1 ijms-22-01603-t001:** MiRNAs involved in CRC pathogenesis.

Function	MiRNAs	Validated Targets	Biological Function and Effect	References
Oncogenic (promotion)	miR-31	RASA1 LATS2 TNS1	Proliferation; migration and invasion; mesenchymal transitions (EMT); Immune cell infiltration	[250,251,252,253,254]
	miR-106b-5p	FAT4	Proliferation, migration and invasion; angiogenesis	[255]
	miR-10b	TP53 P21 PTEN KLF4	Proliferation, Migration, and invasion	[256,257,258]
	miR-100-5p	ATG5 SMAD4	Apoptosis and autophagy; proliferation, migration, and invasion	[259,260]
	miR-124	PLCB1 IQGAP1 ITGA3	Apoptosis and invasion; growth and colony formation ability; anoikis	[261,262,263]
	miR-576-5p	WNT5a AKT3	EMT; proliferation and apoptosis	[264,265,266]
	miR-200a-3p	FOXA1	EMT; migration and invasion; proliferation	[193]
	miR-934	PTEN	Induction of M2 macrophages polarization	[267]
	miR-151-3p	CHL-1	Proliferation and invasion	[268]
	miR-96	AMPKα2	Proliferation and apoptosis	[269,270]
	miR-425-5p	CTNND1	Discriminate KRAS-mutated CRC vs. KRAS-wildtype CRC; growth and metastasis	[250,271]
	miR-224	GSK3β SFRP-2 BTRC Caspase3-7 CPNE8 PHLPP2 NEDD4L LGALSL SH3KBP1 ID3 MAFG PDE8A TMEM9B CHP2 CEP85 FZD5S LC4A4 RIOK3	Proliferation; migration and invasion	[272,273,274,275]
Tumor suppressive (inhibition)	miR-486-5p	PLAGL2 NRP-2	Inhibits of stemness genes; proliferation, migration, and apoptosis; EMT	[276,277,278]
	miR-101	EZH2	Migration	[279]
	miRNA-137	c-MET	Proliferation; growth; migration and invasion	[280]
	miRNA-18a	CDC42 KRAS	Proliferation; migration; apoptosis	[281,282]
	miR-149	IGF2BP1 EPHB3 TCF12 HOXB8 FZD5	Proliferation; migration and invasion; EMT; apoptosis	[273,283,284,285]
	miR-150-5p	SEMA3A SMC2 LRRC58 EPHB2 SLC7A11 PALD1 RPL7L1 GRIN2B HILPDA IL1A INCENP IPO5 SP5 NEK2 MCM10 PRKDC PRPS1 TP53 HMGA2 LRP11 PKP4 JADE3 GINS1 LRG1 CTNNB1	Proliferation; migration and invasion; EMT	[273,286,287,288]
	miR-133b	SMC4 DHRS2 SMYD5 NEBL CCT2 SMC2 CENPF PLK4 PDE10A CHEK1 TMEM123 LRRC58 PAQR4 E2F7 AHCY EPHB4 LRCH1 NUP205 SULF1 NCAPH PATZ1 DCAF13 PHF19 SLCO1B3 HELLS HMGA1 HOXA9 KCNN4 SAMD5 MAD2L1 MKI67 ABCC1 PFAS DDIT4 ATRX PARPBP CDCA8 LRRC36 BRIX1 ZNF280C PRPS2 SLC39A10 PPM1H ALPK3 KIAA1549 PTPRO RFC3 BICD1 XPO4 RANBP17 SIM2 SLC6A6 BMP7 HLTF FSCN1 SOX4 SOX9 SQLE TCF7 TM4SF4 MYRF CA8 SH3TC2 FAM57A PGAP1 TTYH3 REG4 TMPRSS13 ZNRF3 NKD1 STC2 ADAM9 CCNB1 DUSP27 FOXQ1 CD44 LZTS3 CDK1 MELK	Proliferation and metastasis	[273,289]
	miR-133a	MAD2L1 NEBL TMEM123 E2F7 HOXA9 KRT7 SAMD5 PFAS ATRX ZNF280C PRPS2 ALPK3 KIAA1549 PTPRO XPO4 SLC6A6 HLTF FSCN1 SOX4 SQLE TCF7 MYRF FAM57A ZNRF3 STC2 FOXQ1 RHOA	Cell motility	[273,290]
	miR-489	IPO7 SMC2 PAICS PDE10A AP3M2 ZNF618 ESCO2 DUSP4 SLC7A11 DCAF13 OSBPL3 IPO5 ZDHHC9 PKP1 GPCPD1 WDR35 RAN SLC28A3 TFAP4 FZD3 LRP11 PKP4 STC2 ADAM9 CDK1	Proliferation, migration, and invasion	[273,291,292,293]
	miR-139-5p	PTPRU TSPAN5 CDK6 TMEM123 SSX2IP RPL22L1 DPY19L1 CLEC5A HOXA9 ITGA2 MAD2L1 PHKA1 PODXL DNAJC10 DDIT4 ATRX BCOR PSPH PTPRO ROBO2 RYR2 SNTB1 TCF3 TCF12 TGIF1 TOP2A FZD3 PROSER1 ADGRG1 YAP1 Notch1	Proliferation, migration, and invasion	[273,294]
	miR-145-5p	FSCN1 MYRF TUG1 IGF1 CDCA3 RHBDD1	Proliferation and migration; EMT	[273,295,296,297,298]
	miR-222-3p	PTEN SYNCRIP IPO7 SMC2 ZNF618 SH3BP4 OSBPL3 GRB10 AQP3 SBK1 STMN1 MMP1 NAP1L1 DDIT4 SYBU PRPS1 SLC39A10 WDR35 SLC22A3 HLTF TCF12 TFAP2A FZD3 NAA25 AXIN2 CBFB MST3	Migration and invasion	[273,299,300]
	miR-126	mTOR SLUG TWIST VEGFA	Apoptosis and autophagy; angiogenesis; EMT	[301]
	miR-206	hnRNPA1 CCL2 CXCR4 VEGF-A YAP1	Warburg effect; proliferation; migration and invasion	[302,303,304,305]
	miR-330	HMGA2 PFN1 LASP1	Migration and viability; proliferation	[306,307,308]
	miR-144	NRAS SMAD4 ANO1 EZH2	Proliferation; migration and invasion	[309,310,311]
	miR-218-5p	SEC61A1 c-FLIP	Proliferation; migration and invasion; apoptosis	[312,313]
	miR-654-3p	SRC	Proliferation; migration and invasion	[314]
	miR-497-5p	FOXK1 BCL2 ACSL5 AVL9	Proliferation; migration and invasion	[315,316,317,318]

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
