# Peer review of "Portrait of Cancer Stem Cells on Colorectal Cancer: Molecular Biomarkers, Signaling Pathways and miRNAome"

_ijms, 2021, doi:10.3390/ijms22041603_

Round 1

Reviewer 1 Report

This study demonstrates that colorectal cancer stem cells are involved in cancer progress. The suggestion of the monitoring role of miRNAs in biological functions of CSCs. The miRNAome approach is very interesting. The section 2 may be revised to explain how Intestinal Stem Cells drive epithelial homeostasis more in detail. Explanation of the role of miR-147 may be added in section 4. Figure 2 may be revised to indicate each pathway in the figure.

Author Response

  1. The section 2 may be revised to explain how Intestinal Stem Cells drive epithelial homeostasis more in detail.

         Accordingly, with reviewer’s suggestion, we have explained more in details how Intestinal Stem Cells drive epithelial homeostasis (see page 3 lines 129-149 and page 4 lanes 160-170). Moreover, based on the suggestions of reviewer 2, we have shortened the section 2 and 3 eliminating all stem cells and cancer stem cells information do not related to intestine and CRC.

  1. Explanation of the role of miR-147 may be added in section 4.

         According with reviewer suggestion, we describe the role of miR-147 in the colon cancer stem cells in section 4.

(see page 18 lines 816-825)

  1. Figure 2 may be revised to indicate each pathway in the figure.

         According with reviewer suggestion, we indicate the name of each pathway on top in the figure 2, considering the complexity to interpret it.

Finally, we have to inform you that we have reconsidered the title and rewritten it completely based on suggestions of reviewer 2. New title is “Portrait of cancer stem cells on colorectal cancer: molecular biomarkers, signaling pathways and miRNAome”.

Reviewer 2 Report

The manuscript "Portrait of cancer stem cells on colorectal cancer: from pathogenesis to therapies" presented by Andrea Angius et. al., was aimed to review the current research of the molecular, genetic and epigenetic mechanisms linked to miRNAome in the maintenance and regulation of colorectal cancer stem cells (CCSCs), including their relationships with different target genes and signaling pathways. Analysis of these mechanisms can help to identify diagnostic, prognostic, and predictive biomarkers for colorectal cancers and develop CCSCs-targeted therapies.

The review summarizes current published data on molecular biomarkers, signaling pathways and miRNAs that regulate the functions of normal and cancer intestinal stem cells.

The article may be interesting to basic and clinical researchers. The information provided in this manuscript may be useful for further research. The manuscript gives an overview of the latest findings of the field.  The references were used properly.

However, there are some major comments to be addressed.

  1. The text, especially section headings, needs the grammar editing, because many sentences contain errors and ambiguities in the description.
  2. In my opinion, the title of the manuscript does not fully correspond to the content and needs to be clarified. This review discusses the molecular mechanisms that regulate the function of normal and cancer intestinal stem cells, with particular attention to the role of miRNAs. Therefore, such a review describes and discusses a molecular portrait, rather even a miRNA signature of CCSCs.

In addition, the review contains little information and discussion about the clinical application of the obtained data on molecular biomarkers for diagnosis and treatment, therefore the second part of the manuscript title is also unclear - 'from pathogenesis to therapy'.

  1. Sections 2 and 3 (Intestinal Stem Cells drive of epithelial homeostasis and regeneration; and Cancer Stem Cells in Colorectal Cancer) contain redundant (one page) generally known information about the properties of normal and cancer stem cells.

Therefore, these sections should be significantly shortened. These sections should directly begin with information on intestinal stem cells and colorectal cancer stem cells.

  1. Figure 1 (Different origin of cancer stem cells and their roles in tumor progression) is generalized for all types of adult stem cells and does not reflect the origin and properties of colorectal cancer stem cells, which are the main subject of this review. It is necessary to modify it to normal and cancerous intestinal stem cells.
  2. For both Figures 2 and 3, a more detailed Figure legend needs to be provided.

Author Response

  1. The text, especially section headings, needs the grammar editing, because many sentences contain errors and ambiguities in the description.

         According to the Reviewer’s comment, a critical revision of the entire manuscript has been made. The manuscript was completely proofread again by an experienced scholarly writer.

  1. In my opinion, the title of the manuscript does not fully correspond to the content and needs to be clarified. This review discusses the molecular mechanisms that regulate the function of normal and cancer intestinal stem cells, with particular attention to the role of miRNAs. Therefore, such a review describes and discusses a molecular portrait, rather even a miRNA signature of CCSCs. In addition, the review contains little information and discussion about the clinical application of the obtained data on molecular biomarkers for diagnosis and treatment, therefore the second part of the manuscript title is also unclear - 'from pathogenesis to therapy'.

         We are grateful to the reviewer for this suggestion, we have reconsidered the title based on the comment’s reviewer and rewritten it completely: “Portrait of cancer stem cells on colorectal cancer: molecular biomarkers, signaling pathways and miRNAome”.

  1. Sections 2 and 3 (Intestinal Stem Cells drive of epithelial homeostasis and regeneration; and Cancer Stem Cells in Colorectal Cancer) contain redundant (one page) generally known information about the properties of normal and cancer stem cells. Therefore, these sections should be significantly shortened.

         Accordingly, with reviewer’s suggestion, we have shortened the section 2 and 3 eliminating all stem cells and cancer stem cells information do not related to intestine and CRC. The starting of section 3 has been modified (see pages 4-5 lanes 200-218).

Based on the requests of reviewer 1, we have explained more in details how Intestinal Stem Cells drive epithelial homeostasis in section 2.

  1. Figure 1 (Different origin of cancer stem cells and their roles in tumor progression) is generalized for all types of adult stem cells and does not reflect the origin and properties of colorectal cancer stem cells, which are the main subject of this review. It is necessary to modify it to normal and cancerous intestinal stem cells.

         As suggested by the reviewer, we have completely modified Figure 1 improving its specificity for colorectal cancer stem cells.

  1. For both Figures 2 and 3, a more detailed Figure legend needs to be provided.

         We are grateful to the reviewer for this suggestion. We have rewritten and improved all the Figure legends (Figure 1, 2, 3).

Round 2

Reviewer 2 Report

In a revised version of the manuscript, the authors made corrections following reviewer comments. All the reviewer questions were answered. The authors corrected and shortened the text and significantly improved Figure legends. Modified manuscript title is corresponded to the content. New Figure 1 (Origin of colorectal cancer stem cells and their role in tumor progression) now reflects the origin and properties of colorectal cancer stem cells. Therefore, in my mind, revised manuscript satisfies all the requirements for publication.

Minor comments:

Figure 3: E-CADERIN - needs  a correction to E-CADHERIN

Author Response

  1. Figure 3: E-CADERIN - needs  a correction to E-CADHERIN

We are grateful to the reviewer for this suggestion. We have corrected the mistake related to the name of E-Cadherin gene in figure 3.